# Long-Term Clinical Outcome in Systemic Lupus Erythematosus Patients Followed for More Than 20 Years: The Milan Systemic Lupus Erythematosus Consortium (SMiLE) Cohort

**DOI:** 10.3390/jcm11133587

**Published:** 2022-06-22

**Authors:** Maria Gerosa, Lorenzo Beretta, Giuseppe Alvise Ramirez, Enrica Bozzolo, Martina Cornalba, Chiara Bellocchi, Lorenza Maria Argolini, Luca Moroni, Nicola Farina, Giulia Segatto, Lorenzo Dagna, Roberto Caporali

**Affiliations:** 1Research Centre for Adult and Pediatric Rheumatic Diseases, Department of Clinical Sciences and Community Health, University of Milan, 20129 Milan, Italy; roberto.caporali@unimi.it; 2ASST Pini CTO, Lupus Clinic, Division of Clinical Rheumatology, 20122 Milan, Italy; martinacornalba2@gmail.com (M.C.); lorenza.argolini@hotmail.it (L.M.A.); 3Fondazione IRCCS Ca’ Granda Ospedale Maggiore Policlinico di Milano, Referral Centre for Systemic Autoimmune Diseases, 20122 Milan, Italy; lorberimm@hotmail.com (L.B.); chiara.bellocchi@unimi.it (C.B.); giulia.segatto@unimi.it (G.S.); 4IRCCS Ospedale San Raffaele, Unit of Immunology, Rheumatology, Allergy and Rare Diseases, 20132 Milan, Italy; ramirez.giuseppealvise@hsr.it (G.A.R.); bozzolo.enrica@hsr.it (E.B.); moroni.luca@hsr.it (L.M.); farina.nicola@hsr.it (N.F.); dagna.lorenzo@hsr.it (L.D.)

**Keywords:** lupus, flare, damage, long disease duration, trajectories, remission, low disease activity

## Abstract

Tackling active disease to prevent damage accrual constitutes a major goal in the management of patients with systemic lupus erythematosus (SLE). Patients with early onset disease or in the early phase of the disease course are at increased risk of developing severe manifestations and subsequent damage accrual, while less is known about the course of the disease in the long term. To address this issue, we performed a multicentre retrospective observational study focused on patients living with SLE for at least 20 years and determined their disease status at 15 and 20 years after onset and at their last clinical evaluation. Disease activity was measured through the British Isles Lupus Assessment Group (BILAG) tool and late flares were defined as worsening in one or more BILAG domains after 20 years of disease. Remission was classified according to attainment of lupus low-disease-activity state (LLDAS) criteria or the Definitions Of Remission In SLE (DORIS) parameters. Damage was quantitated through the Systemic Lupus Erythematosus International Collaborating Clinics/American College of Rheumatology damage index (SLICC/ACR-DI). LLAS/DORIS remission prevalence steadily increased over time. In total, 84 patients had a late flare and 88 had late damage accrual. Lack of LLDAS/DORIS remission status at the 20 year timepoint (*p* = 0.0026 and *p* = 0.0337, respectively), prednisone dose ≥ 7.5 mg (*p* = 9.17 × 10^−5^) or active serology (either dsDNA binding, low complement or both; *p* = 0.001) were all associated with increased late flare risk. Late flares, in turn, heralded the development of late damage (*p* = 2.7 × 10^−5^). These data suggest that patients with longstanding SLE are frequently in remission but still at risk of disease flares and eventual damage accrual, suggesting the need for tailored monitoring and therapeutic approaches aiming at effective immunomodulation besides immunosuppression, at least by means of steroids.

## 1. Introduction

Systemic lupus erythematosus (SLE) is a chronic autoimmune disease of unknown aetiology, characterized by a broad spectrum of clinical presentations, and linked to the production of autoantibodies leading to inflammation with multi-organ involvement. Over time, SLE morbidity may be affected not only by exacerbations, but also by progressive organ damage, either related to SLE, treatments or comorbidities [1,2,3,4]. The treat-to-target (T2T) therapeutic approach, aiming at “remission of systemic symptoms and organ manifestations” is founded on this notion, with the perspective of minimising patient disability and improving long-term survival [5,6,7]. In the past, different definitions of remission were proposed [1,8], but none were universally accepted, hindering the selection of an appropriate outcome measure for any T2T strategy. Starting from 2016, the Definitions Of Remission In SLE (DORIS) Initiative provided a framework for defining remission in SLE [2,9]. Then, the task force performed a thorough revision of accumulating evidence, information and data to reach the final recommendations for a definition of remission in SLE in 2021 [10].

While remission is an achievable outcome, it may seldom be reached or sustained with the current therapies and thus the development of a more attainable target associated with reduced damage accrual protection has been advocated. The Asia Pacific Lupus Collaboration (APLC) performed a series of studies to validate the so-called Lupus Low Disease Activity State (LLDAS). The LLDAS is a composite score evaluating five different aspects of the disease, and has proven to be a valuable alternative for the implementation of treat-to-target therapeutic strategies. It has been demonstrated that patients in persistent LLDAS show a significantly lower frequency of disease flare-ups and lower accumulation of organ damage [11,12].

With the introduction of new therapies, there has been considerable improvement in the survival of individuals with SLE. In contrast, patients living longer with the disease may present chronic organ damage and disability as a result of persistent disease activity and/or treatment side effects [13,14,15].

Few studies are available on patients affected by long-standing SLE. According to these, skin and joint involvement are associated with a lower likelihood of achieving LLDAS or remission, probably because of the still suboptimal control of these symptoms based on current therapies [16,17].

Based on the previous considerations, the objectives of this study are: the evaluation of the proportion of patients achieving remission according to DORIS definitions or low disease activity, LLDAS, in a multicentre cohort of SLE patients with disease duration of more than 20 years; the identification of valid prognostic markers to support such remission or LLDAS, and prevent possible disease flare-ups; the estimation of the effect of prolonged remission or LLDAS on damage accumulation in patients with long-standing disease.

## 2. Materials and Methods

### 2.1. Patients

The Milan Systemic Lupus Erythematosus Consortium (SMiLE) cohort is a longitudinal observational cohort of SLE patients regularly followed at three rheumatology tertiary centres in Milan: Lupus Clinic of the Clinical Rheumatology Unit of ASST Pini-CTO, the Referral Center for Systemic Autoimmune Diseases of Fondazione Ca’ Granda IRCCS Policlinico and the Lupus Clinic of the Unit of Immunology, Rheumatology, Allergy and Rare Diseases at IRCCS Ospedale San Raffaele. The cohort was created with the aim to better characterise the biological and clinical features of SLE patients and includes all patients fulfilling the 1997 American College of Rheumatology revised classification criteria or the 2019 SLICC/ACR revised criteria for the diagnosis of SLE. The protocol was approved by the IRCCS San Raffaele Hospital Ethics Committee and the Comitato Etico Milano Area 2 (approval no. 0002450/2020) (for both the “Pini” and “Policlinico” hospitals). When entering the cohort, all patients signed an informed consent form. 

Within this population, we selected all the patients with a disease duration ≥20 years. Medical records of patients were retrospectively evaluated and clinical and laboratory information were collected at baseline, at 15 and 20 years from the first visit and at the time of the last observation. 

### 2.2. Outcome Measures

Disease activity, severity of organ involvement and flares were assessed by the British Isles Lupus Assessment Group 2004 (BILAG-2004) index [18,19,20]. In addition, the Systemic lupus erythematosus disease activity index 2000 (SLEDAI-2K) [21] and the Physician Global Assessment (PGA) 0–3 scale [22] were also scored. Remission was defined according to the DORIS final recommendations [10] and namely as a clinical SLEDAI (cSLEDAI) = 0, a PGA < 0.5, irrespective of serology; patient may be on antimalarials, low-dose glucocorticoids (prednisone or equivalents ≤ 5 mg/day), and/or stable immunosuppressives including biologics. Low-disease-activity state (LLDAS) was defined as recently reported by Franklyn et al. [12], that is, an SLEDAI-2K ≤ 4, no disease activity in major organ systems (renal, central nervous system [CNS], cardiopulmonary, vascular, fever), no occurrence of haemolytic anaemia or gastrointestinal activity, no new disease activity in relation to previous evaluations, a PGA ≤ 1; patient may be on low-dose glucocorticoids (prednisone or equivalents) ≤ 7.5 mg/day), standard maintenance dose of well-tolerated, approved, immunosuppressive drug and biologic therapies, excluding the investigational drugs. Estimated damage accumulation was calculated using the Systemic Lupus International Collaborating Clinics/American College of Rheumatology Damage Index (SLICC/ACR-DI) [23].

Low complement was defined as C3 and/or C4 levels below the reference value. Anti-dsDNA positivity was considered when values where above twice the threshold value, while and anti-PL positivity was confirmed when at least one among aCL, anti-b2GPI and LAC was positive in 2 determinations 12 weeks apart.

### 2.3. Statistical Analysis

For descriptive statistics, continuous variables were summarized as mean ± standard deviation, except for skewed data that were described as median (interquartile range). The chi-squared test was used to compare categorical variables from 2 × 2 contingency tables.

Survival analysis was conducted to estimate the time to the first flare occurring after the 20th year of disease. The analysis was restricted to the 20–30th year interval to ensure that at least 20% of cases were available at the end of observation and to provide reliable survival estimates. Time-to-event analysis was conducted via Cox regression with the estimation of hazard ratios (HR) and relative 95% confidence intervals (CI95).

## 3. Results

### 3.1. Clinical Features at Enrolment and at Study End

Long-term data were available for 221 patients with a mean follow-up of 28.5 ± 6.6 years from diagnosis (Table 1). Musculoskeletal and mucocutaneous involvements were the most prevalent manifestations in patient history, and nearly half of the patients (*n* = 106) experienced lupus nephritis. Renal involvement was the presenting manifestation in 32.2% of cases and usually occurred early in the medical history (81.3% of times within 10 years, 91.6% of times within 15 years). The vast majority of patients (*n* = 202, 91.4%) were treated with hydroxychloroquine and 172 subjects (77.8%) were exposed to conventional immunosuppressants. At the end of the observation (28.5 ± 6.6 years from diagnosis), 129 patients were both in LLDAS and DORIS remission and 41 patients were neither in LLDAS or DORIS remission. The two remission classifications were significantly associated (χ^2^ = 63.940; *p* < 0.0001). A total of 172 patients (77.8%) had accumulated one or more SLICC/ACR-DI items with a mean of 1.7 ± 1.73 items/patient. Cataract was the most prevalent damage item (*n* = 33, 14.9%), followed by erosive/deforming arthritis (*n* = 26, 11.8%) and osteoporosis with fractures (*n* = 25, 11.3%), as detailed in Appendix A. 

### 3.2. Changes in Disease Activity and Damage Accrual over Time

The chance of being in LLDAS or in remission, according to DORIS definition, increased over time. Nonetheless, chronic damage according to the SLICC/ACR-DI, was also higher when patients were observed at later timepoints (Table 2). BILAG scores for each disease domain were collected at each timepoint and are reported in Appendix A. Disease flare rates after 20 years of follow up were calculated based on worsening BILAG scores in one or more domains. Eighty-four subjects (38.9%) had one or more flares, yielding a 10-year flare risk of nearly 50% (Figure 1). Most flares were experienced in the musculoskeletal domain, as shown in Table 3. In 10 cases (11.9%), a BILAG A flare was preceded by a BILAG B score, in 19 cases (22.6%) a BILAG A or B flare was preceded by a BILAG C score, by a D score in 41 (48.8%) and by an E score in 14 (16.7%). Pre-flare serological status was altered in 46 (55.4%) patients: 40 had (47.6%) low complement, 26 (30.9%) had increased dsDNA binding, and 18 (21.4%) had a negative serological status. Regarding therapy before flare, 21 subjects (25%) were treated with prednisone >5 mg/day, 45 (53.7%) with hydroxychloroquine and 39 (46.4%) with immunosuppressants. In total, 88 patients of 216 with available data (40.7%) accrued additional damage items after the 20-year timepoint. 

### 3.3. Factors Associated with Late Flares and Damage Accrual 

Patients in LLDAS at the 20-year timepoint had nearly half the risk of a flare within the following ten years compared to patients who were not in LLDAS (HR = 0.487, CI95 = 0.305–0.778, *p* = 0.0026; Figure 2). Similar results were observed considering the attainment of DORIS remission at 20 years of disease (HR = 0.611, CI95 = 0.338–0.963, *p* = 0.0337). Patients with low complement and increased dsDNA binding at pre-flare evaluation had a higher 10-year risk of flare compared to patients with a fully negative serology (HR = 2.645, CI95 = 1.478–4.73, *p* = 0.001), while the occurrence of either serological alteration was not associated with any risk (Figure 3). No other clinical features at the 20-year timepoint were associated with eventual flares. Patients taking prednisone or equivalent at a dosage ≥ 7.5 mg/day before flare had a higher flare risk in the subsequent follow up compared to patients with a lower-dose or off steroids (HR = 2.684, CI95 = 1.637–4.403, *p* = 9.17 × 10^−5^). 

Late flares (that is, occurring after 20 years of follow up, N = 52/88, 59%) were more frequent among patients with progressing SLICC/ACR-DI (N = 52/88, 59.0%) than among patients who did not accrue additional damage after 20 years of follow up (N = 41/128, 32.0%; OR = 3.411, CI95 = 1.90–6.12, *p* = 2.7 × 10^−5^).

## 4. Discussion

In this multicentre study, we analysed the trajectories of a relatively large cohort of patients with SLE with long disease duration. We found that most patients have low to no disease activity when observed from the 15th year of disease on, and that the proportion of patients in remission increases over time. However, late damage accrual was also experienced by 40% of patients, possibly with a linear trend. Patients who were not in remission or LLDAS at the 20-year timepoint had a higher risk of late flares, which in turn were associated with damage accrual. Consistently, patients needing higher prednisone doses at pre-flare evaluation had higher chances of developing a flare than patients who had been able to de-escalate or discontinue their treatment with prednisone. Increased dsDNA binding and low complement were also associated with the development of late flares.

Tackling disease progression from persisting activity to damage and disability constitutes a major goal for patients with SLE [3]. Damage accrual has in fact also been associated with higher mortality rates, besides its impact on quality of life [7,24,25,26,27]. Patients with early-onset disease and patients in the early phase of their disease course have consistently been reported to experience more severe manifestations and accrue damage more rapidly [28,29,30]. Damage accrual is predicted to grow linearly at least during the first two decades of disease duration, while less is currently known for later timepoints [27,29,31]. Our data suggest that this linear trend might progress even in very late phases of the disease, which prompts the identification of factors that may modulate this unfavourable course. Consistent with previous studies, we observed that the achievement of a stable remission might protect from late flares and eventually damage [15,17,28,32,33]. Notably, we also confirmed that LLDAS largely overlaps with clinical remission [15] and might constitute a feasible treatment endpoint both in clinical trials and routine rheumatology practice [26,34,35]. In line with other reports, we also found that the musculoskeletal system was most frequently involved in late flares and more prone to be involved in damage accrual [27,29]. In fact, deforming/erosive arthritis and osteoporotic fractures were among the most frequent SLICC/ACR-DI items in our cohort. The relatively lower rate of avascular necrosis compared to other studies might be attributed to ethnic or geographic factors affecting vitamin D metabolism [29,36]. Indeed, high rates of cataract development were also observed in our cohort along with osteoporosis fractures, cardiovascular and retinal complications, possibly suggesting a role of corticosteroid-related mechanisms in contributing to the accrual of damage [37,38]. Taking these data together with the association of high-dose corticosteroid treatments with late flares and that of late flares with damage suggests that distinct treatment approaches might apply to patients at distinct stages of the disease. Immunomodulation through antimalarials and belimumab has been associated with slower damage progression [37,39,40] and might be favoured in older patients with longstanding disease over maintenance immunosuppression. Nonetheless, data from the literature also suggest that immunomodulation is most effective before the onset of initial damage [41], damage itself being a risk factor for further SLICC/ACR-DI progression [7]. The role of immunomodulatory treatment in minimising the effects of deranged B-cell responses in SLE might also be consistent with the predictive role of active combined increased dsDNA binding and low complement towards late flares.

The results of this study suggest that late flares herald damage accrual in patients with longstanding disease and might correlate with higher doses of corticosteroids. This evidence should however be put into the context of some study limitations. First, data regarding the preceding disease history were incomplete, as digital clinical records were only recently introduced into our clinical setting and hard copies of older documents were not available for all subjects. Second, data collection was planned at discrete timepoints rather than visit-by-visit, preventing comprehensive tracking of disease fluctuations over time. Third, treatment features were not homogeneous among subjects, possibly reflecting the evolution of lupus care during recent decades but introducing potential confounders in terms of deflection of disease activity and damage accrual trajectories. Fourth, we focused on a relatively limited number of clinical variables without complementary assessment of potential biomarkers, preventing the development of further patient stratification by pathophysiological or phenotype features.

## 5. Conclusions

Notwithstanding these limitations, our data provide novel hints regarding disease- and treatment-related morbidity in patients with longstanding SLE. Even patients with longer distance from disease onset, especially those with persistent active serology, are at risk of developing disease flares, which in turn constitute a risk factor for late damage accrual. Taken together, our data support the need for closer monitoring of these patients, even in the long term.

## Figures and Tables

**Figure 1 jcm-11-03587-f001:**
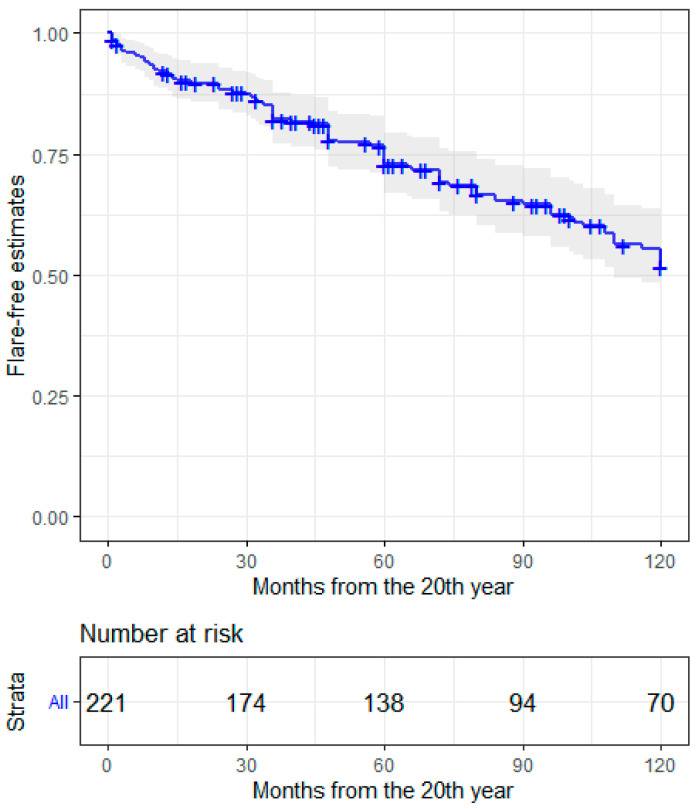
Flare-free estimates in long-term SLE patients up to 10 years from the 20th year of disease.

**Figure 2 jcm-11-03587-f002:**
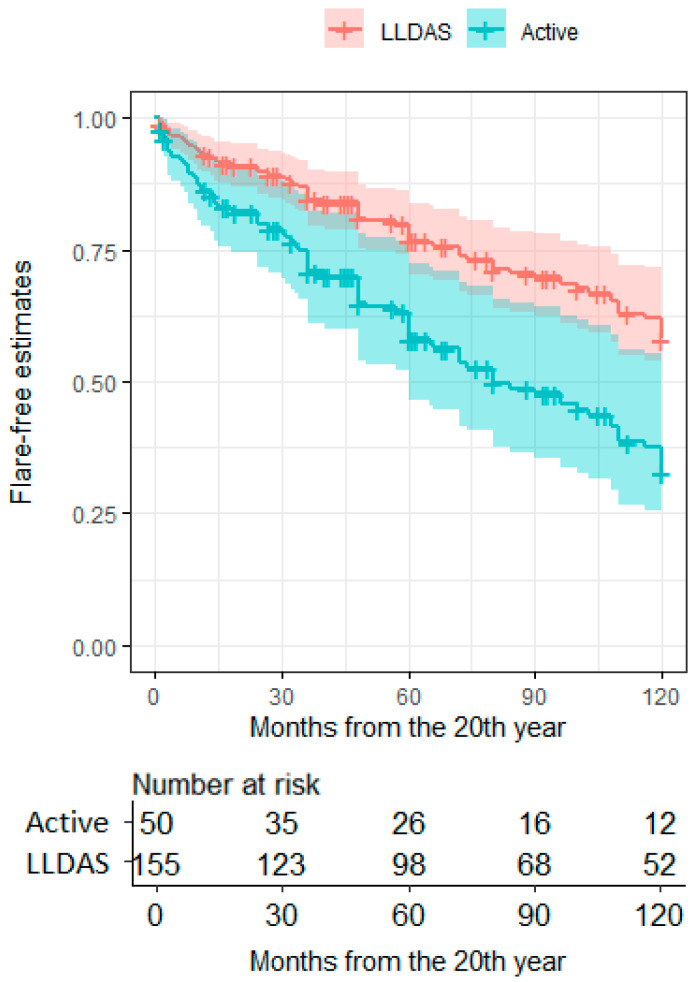
Ten-year flare-free estimates in long-term SLE patients according to the LLDAS status at 20 years. Flare-free survival estimates after the 20th year of disease according to the activity state at the 20th year. LLDAS, lupus low-disease-activity state; Active, absence of LLDAS (active disease).

**Figure 3 jcm-11-03587-f003:**
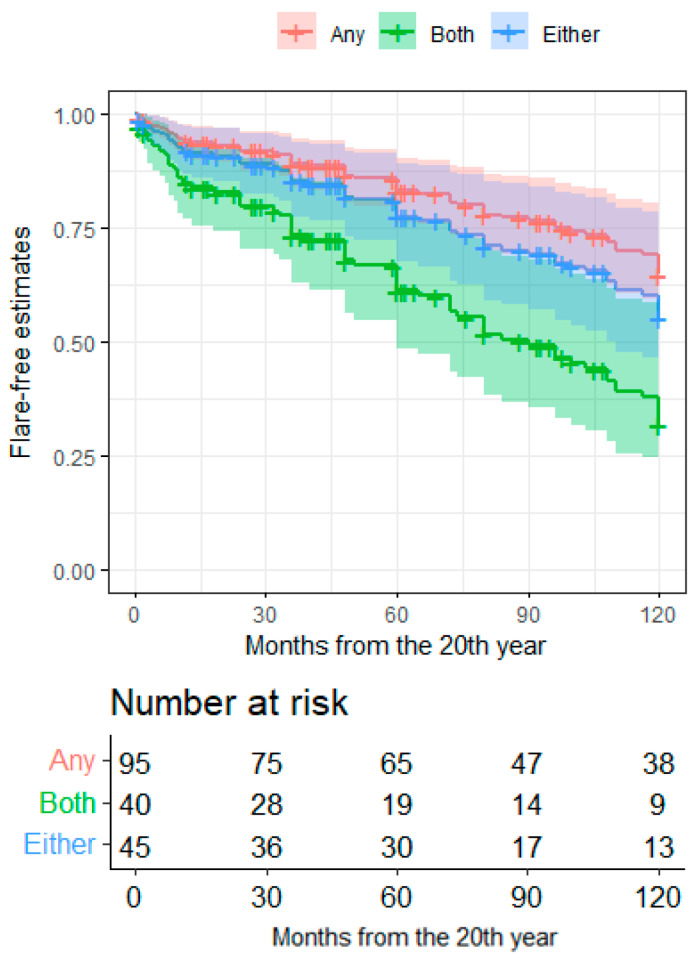
Ten-year flare-free estimates in long-term SLE patients according to serology at 20th years. Flare-free survival estimates after the 20th year of disease according to the serological status at the 20th year. Both—presence of both serological alterations (low dsDNA AND hypocomplementemia); Either—presence of either serological alterations (low dsDNA OR hypocomplementemia); Any—absence of any serological alteration.

**Table 1 jcm-11-03587-t001:** Clinical, laboratory and demographic characteristics.

Variable	*n* = 221
Female, *n* (%)	198 (89.6%)
Age at diagnosis, years	25.6 ± 10.6
Follow-up, years	28.5 ± 6.6
Serology, *n* (%)	
Anti-dsDNA	177 (80.1%)
Anti-Sm	33 (14.9%)
aPL	98 (44.3%)
Low complement	181 (81.9%)
Clinical features ever, *n* (%)	
Musculoskeletal involvement	189 (85.5%)
Mucocutaneous involvement	180 (81.4%)
Renal involvement	106 (48.8%)
Neuropsychiatric SLE	49 (22.2%)
Cardiopulmonary involvement	75 (33.9%)
Haematological involvement	138 (62.4%)
Constitutional symptoms	168 (76%)
Gastrointestinal involvement	11 (5%)
Ophthalmic involvement	19 (8.6%)
Treatment ever, *n* (%)	
Hydroxychloroquine	202 (91.4%)
Prednisone ≥5 mg	221 (100%)
Methotrexate	51 (23.1%)
Mycophenolate mofetil/mycophenolic acid	72 (32.6%)
Azathioprine	113 (51.1%)
Cyclosporine	49 (22.2%)
Cyclophosphamide	72 (32.6%)
High-dose intravenous steroids	117 (52.9%)
Rituximab	14 (6.3%)
Belimumab	25 (11.3%)

ds-DNA: double strain-DNA; anti-Sm: anti-Smith; aPL: anti-phosholipid antibodies; SLE: Systemic Lupus Erythematosus.

**Table 2 jcm-11-03587-t002:** LLDAS, remission ad SLICC/ACR-DI indexes at long-term endpoints.

Status	15 Years (*n* = 199)	20 Years (*n* = 205)	Last Observation (*n* = 221)
LLDAS	137 (68.8%)	155 (75.6%)	177 (80%)
DORIS remission	107 (53.8%)	115 (58%) *	132 (59.7%)
SLICC/ACR-DI	0.57 ± 1.04 **	0.89 ± 1.42 ***	1.57 ± 1.9

Full dataset available at the last observation; exploratory analysis on available data at 15 and 20 years from diagnosis. Data from: * 183 patients; ** 198 patients; *** 216 patients. LLDAS: Lupus low disease activity state; DORIS: Definitions Of Remission In SLE; SLICC/ACR-DI: Systemic Lupus Erythematosus International Collaborating Clinics/American College of Rheumatology damage index.

**Table 3 jcm-11-03587-t003:** Clinical features of patients with flares after 20 years of follow up by BILAG domain.

Organ/Apparatus	Flares (*n* = 84)
Musculoskeletal	38 (45.2%)
Mucocutaneous	15 (17.9%)
Renal	9 (10.7%)
Neuropsychiatric SLE	6 (7.1%)
Cardiopulmonary	3 (3.6%)
Haematological	6 (7.1%)
Constitutional	2 (2.4%)
Gastrointestinal	3 (3.6%)

SLE: Systemic Lupus Erythematosus.

## Data Availability

Data are available upon request from the corresponding author. The data are not publicly available due to privacy reasons.

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
