# Peer review of "Long-Term Clinical Outcome in Systemic Lupus Erythematosus Patients Followed for More Than 20 Years: The Milan Systemic Lupus Erythematosus Consortium (SMiLE) Cohort"

_jcm, 2022, doi:10.3390/jcm11133587_

Round 1
Reviewer 1 Report
Maria Gerosa et al did a great job. They focused on the long term clinical outcome in SLE patients, and found disease status, GC dose, and serological pattern associated with late flare and damage accrual. I will suggest the authors further analysis the relationship between comorbidities, infection, GC accumulated dose, number of past relapses and late flare.
Author Response
We are thankful to the reviewer for his/her appreciation of our work and for the useful suggestion. Unfortunately data for the study were manually retrieved from medical records, some of them dating back to the ‘80s or ‘90s and were not collected from a structured database. For this reason not all the clinical data are readily available and should de novo collected when available (a number of missing data are foreseen); the systematic analysis of the suggested relationships is therefore not possible.
Reviewer 2 Report
This paper describes the long-term outcome of SLE patients with disease duration of more than 20 years. The topic is of interest, available data on it limited. The data overall are well and concisely presented. I have mostly minor concerns with the aim to improve the quality of the manuscript:
- Please define the serology in the methods, in particular with regard to ‘low complement’ (what does it mean ? low C3 ? low C4 ? low CH50 ? any of the three ?)
- I am missing more detailed figure legends. For Example: Please formulate more clearly what you mean with “either”, “any” etc. . How did the authors define “active” disease ? Absence of remission ??
- The conclusion for me should be rather part of the discussion the way as it is written now. Either skip an additional conclusion part or try to find a better way to sum up and draw a conclusion of the findings
Author Response
This paper describes the long-term outcome of SLE patients with disease duration of more than 20 years. The topic is of interest, available data on it limited. The data overall are well and concisely presented. I have mostly minor concerns with the aim to improve the quality of the manuscript:
- Please define the serology in the methods, in particular with regard to ‘low complement’ (what does it mean ? low C3 ? low C4 ? low CH50 ? any of the three ?)
We added a sentence in the methods explaining the definition of low complement and autoantibody positivity.
- I am missing more detailed figure legends. For Example: Please formulate more clearly what you mean with “either”, “any” etc. . How did the authors define “active” disease? Absence of remission ??
Legends to Figures 2 and 3 have now been added
- The conclusion for me should be rather part of the discussion the way as it is written now. Either skip an additional conclusion part or try to find a better way to sum up and draw a conclusion of the findings
The concluding remarks, required as per journal guidelines, have now been shortened and partially incorporated in the main discussion text.
Reviewer 3 Report
The authors presented the long-term clinical outcome in systemic lupus erythematosus by the retrospective study. This article appears to be important, especially as it shows some factors pertaining to remission in the patients followed more than 20 years. However, there are some concerns to be published.
1. Did the patients receive oral corticosteroids? If so, the number (%) of patients who were treated with oral corticosteroids should be shown in the table 1.
2. The maximum value of y-axis should be 100 (%) in the supplemental figure 1. The tables under the bar graphs look unnecessary.
Author Response
The authors presented the long-term clinical outcome in systemic lupus erythematosus by the retrospective study. This article appears to be important, especially as it shows some factors pertaining to remission in the patients followed more than 20 years. However, there are some concerns to be published.
- Did the patients receive oral corticosteroids? If so, the number (%) of patients who were treated with oral corticosteroids should be shown in the table 1.
We did not include this information in the text because all the patients received oral steroids at onset or during the course of the disease. We add this data in table 1.
- The maximum value of y-axis should be 100 (%) in the supplemental figure 1. The tables under the bar graphs look unnecessary.
The figure has been corrected as suggested
Round 2
Reviewer 1 Report
I have no other comment on the article. The paper can be considered for publication.